# Roles of Carbonic Anhydrases and Carbonic Anhydrase Related Proteins in Zebrafish

**DOI:** 10.3390/ijms23084342

**Published:** 2022-04-14

**Authors:** Ashok Aspatwar, Leo Syrjänen, Seppo Parkkila

**Affiliations:** Faculty of Medicine and Health Technology, Tampere University, 33520 Tampere, Finland; leo.syrjanen@tuni.fi (L.S.); seppo.parkkila@tuni.fi (S.P.)

**Keywords:** carbonic anhydrase, carbonic anhydrase related proteins, acid-base balance, ion transport, pH regulation, motor coordination, zebrafish

## Abstract

During recent decades, zebrafish (*Danio rerio*) have become one of the most important model organisms in which to study different physiological and biological phenomena. The research field of carbonic anhydrases (CAs) and carbonic anhydrase related proteins (CARPs) is not an exception to this. The best-known function of CAs is the regulation of acid–base balance. However, studies performed with zebrafish, among others, have revealed important roles for these proteins in many other physiological processes, some of which had not yet been predicted in the light of previous studies and suggestions. Examples include roles in zebrafish pigmentation as well as motor coordination. Disruption of the function of these proteins may generate lethal outcomes. In this review, we summarize the current knowledge of CA-related studies performed in zebrafish from 1993–2021 that was obtained from PubMed search.

## 1. Introduction

Carbonic anhydrases (CAs, EC 4.2.1.1) are metalloenzymes that catalyze the reversible hydration reaction of carbon dioxide according to the following equation of bicarbonate equilibrium: CO_2_ + H_2_O ⇋ HCO_3_^−^ + H^+^ [1]. This reaction catalyzed by CAs is fundamental in the regulation of the acid–base balance in living organisms. In addition, this reaction is involved in many physiological processes such as gluconeogenesis and ureagenesis, and it also helps to remove carbon dioxide out of tissues. The active site of α-CAs contains a zinc ion coordinated by three histidine residues, but in other CA-classes other metal cofactors can also be found; for example, cadmium in the ζ-CAs and iron or cobalt in γ-CAs [2,3].

CAs are grouped into eight evolutionarily distinct classes: α, β, γ, δ, ζ, η, θ, and ι [4,5]. α-CAs belonging to CA class are the best characterized, and the enzymes are present in many prokaryotic and eukaryotic organisms. According to bioinformatic analysis, β-CAs seems to be the class with the widest distribution. Enzymes of β-class are expressed in many prokaryotes and eukaryotes, including plants, fungi, protozoans, nematodes, and arthropods [6,7,8,9,10]. However, vertebrate species possess only α-CAs. In total, 13 enzymatically active α-CA isozymes have been discovered in mammals [11], and in the case of zebrafish, there are a total of 20 α-CA isozymes (https://zfin.org/) (accessed on 8 March 2022) [12]. In addition to catalytically active proteins, some proteins are catalytically inactive and are referred to as carbonic anhydrase related proteins (CARPs) [13].

Zebrafish (*Danio rerio*) is a tropical freshwater fish. Being a vertebrate model organism, its anatomical structures, as well as the physiological functions of most of its organs, are analogous to the features of humans. Genetic analysis has revealed that 69% of zebrafish genes have at least one human ortholog. On the other hand, 71% of human genes have at least one zebrafish ortholog [14]. Even more importantly, 82% of disease-causing genes in humans have an ortholog in zebrafish. An adult female produces approximately 200 eggs per week and has a generation time of about three months, that makes the generation of knockout models, for example, relatively fast [15]. In addition, it is also affordable to maintain a large number of zebrafish in a relatively small laboratory space. Zebrafish embryos are transparent and develop rapidly which enables simple observation of developmental processes [15]. Zebrafish are also compatible with the paradigms of drug discovery; the small size and transparency of the embryos makes them amenable for the automation necessary in high-throughput screening of chemical compounds. Furthermore, the use of zebrafish as a model organism is usually considered more ethical compared to higher vertebrates (e.g., mouse). 

There are a number of different techniques used to create transgenic or and knockout zebrafish model organisms. These tools include morpholino oligonucleotides or MOs [16], TILLING [17], zinc-finger nucleases (or ZFNs) [18], CRISPR/Cas [19,20], TALENs [21] and Tol2 transposons combined with bacterial artificial chromosomes [22]. In addition, the zebrafish research community has created a large number of transgenic, knockdown and knockout models that have been maintained for studying the functions of different genes. Many of these are commercially available.

In the case of CAs, morpholino oligonucleotides and CRISPR/Cas technologies have been used to create knockdown and knockout zebrafish model organisms for CA-genes 5, 6, 8, 10a, 10b, 14 and 17a. This has opened novel possibilities to study the physiological roles of CAs, both in adult zebrafish and during embryonic development. In this review, we cover the current status (Table 1) of CA research performed in zebrafish. 

## 2. Carbonic Anhydrase 2a Plays a Role in Ion Transport and Gas Exchange 

The zebrafish *ca2* gene was first isolated and characterized by Linser et al. in 1997, both at mRNA and protein level [23]. The ca2a mRNA contains 1537 base pairs and the deduced amino acid sequence includes 260 amino acids. The molecular weight of isolated protein was 29 kDa and it showed high enzymatic activity. The *ca2a* gene was suggested to be diverged after the branching of the CA 5 and CA 7 genes. Among the zebrafish CAs, ca2 was the first characterized CA gene from teleost. Studies have shown that the CA II is highly identical to the ca2 from mammals in primary and tertiary structures and genomic organization and syntenic map [24].

Real-time RT-PCR expression analysis showed that the *ca2a* mRNA is expressed during the early stage of zebrafish development. In situ hybridization studies showed that the *ca2* transcript is expressed abundantly in a specific group of cells in the gills and skin and immunohistochemical studies showed its presence in and H ^+^ -ATPase-rich (HR) cells [25]. Expression analysis by immunostaining showed prominent staining of ca2 and Na^+^-K^+^-ATPase in ion-transporting cell (ionocytes) and hair cells, indicating its role in transepithelial transport of HCO^3−^ across the membranous labyrinth into the endolymph [26]. The zebrafish cytosolic *ca2* is also widely expressed in RBCs [39]. Presence of *ca2* mRNA is also expressed in in apical Na^+^/H^+^ exchanger 3b (Nhe3b) [25,27]. In zebrafish, the acid-base balance is believed to be modulated via solute carrier family 26 (SLC26), which is dependent on the cytosolic CAs [40]. The ionocytes are important sites of Na^+^ uptake that are mediated by Nhe2b [41]. During embryonic development, abundant *ca2* was seen in the Na^+^ accumulating mitochondria-rich (MR) cells on the yolk sac, both at mRNA and protein level [27,28].

In an immunostaining study, Miller et al. showed—using immunostaining—that neuroepithelial cells (NECs) are enriched with *ca2* at 4 days post-fertilization (dpf) [29]. The NECs are homologous to mammalian carotid body glomus cells, and in zebrafish CO_2_ is believed to be detected by these cells and hypothesized that inhibition of CA activity would interfere with the downstream responses. Indeed, a study showed that the cardiac response to hypercapnia (buildup of carbon dioxide in your bloodstream) was reduced in fish at 5 dpf exposed to acetazolamide, a CA inhibitor, and in fish knocked down for *ca2* [29].

Knockdown of *ca2* leads to a decrease in H^+^ activity and increase in Na^+^ uptake at 96 hpf with concomitant upregulation of znhe3b and downregulation of zatp61a (H^+^-ATPase A-subunit expression) [25]. Indeed, *ca2* plays a role in Na^+^ uptake and acid-base regulation mechanisms in zebrafish HR cells [25]. The ca2a enzyme was found to be present in cytoplasm and the suppression of *ca2* resulted in a decrease in Na^+^ accumulation in H-MRCs. An in situ proximity ligation assay demonstrated a very close association of *ca2a*, Nhe3b, and Rhcg1 ammonia transporter in H-MRC, suggesting that the *Ca2a*, and Rhcg1 play a key role in Na^+^ uptake by forming a transport metabolon with Nhe3b [27]. The rates of CO_2_ excretion increased approximately 15-fold from 24 to 48 hpf, whereas the rates of O_2_ uptake increased only 6.7-fold over the same period, indicating a relative stimulation of CO_2_ excretion over O_2_ uptake. Treatment of 48 hpf larvae with the CA inhibitor acetazolamide resulted in CO_2_ excretion rates that were 52% of the value in control larvae, a significant difference that occurred in the absence of any effect on O_2_ uptake. Measurement of gas transfer rates at 48 hpf indicated that CA knockdown caused a significant inhibition of CO_2_ excretion compared to O_2_ uptake. These results suggest that between 24 hpf and 48 hpf, developing zebrafish begin to rely on ca2a to meet requirements for increased CO_2_ excretion [30].

Studies have shown that knockdown/knockout of *ca2* using morpholinos and CRISPR/Cas9 results in an increase in whole body Na^+^ uptake in zebrafish larvae [25,31]. In addition, a recent study revealed that complete lack of *ca2* is lethal for zebrafish unlike human CAII deficiency [1,31,42]. The zebrafish larvae knocked out for *ca2* resulted in 100% mortality by 19 dpf [31]. Morphologically, the homozygous mutant larvae knocked out for *ca2* exhibited shorter body length as early as 4 dpf. Interestingly, the heterozygous larvae showed intermediate phenotype, suggesting a partial preservation in the presence of one wildtype allele. The *ca2*^−/−^ zebrafish larvae were characterized by an over or underinflated swim bladder. The study demonstrated that knockout of *ca2* stimulated Na^+^ uptake and reduced Cl^−^. In addition, knockout created by CRISPR/Cas9, morpholino knockdown and treatment with ethoxzolamide all suggested that *ca2* is essential for uptake of Cl^−^ zebrafish [31]. 

## 3. Carbonic Anhydrase 2b Plays a Role in CO_2_ Excretion

Expression analysis using whole-mount in situ hybridization showed *ca2b* is specially expressed in the skin and gills of the developing embryos [25]. The tissue specific expression analyses showed ubiquitous presence of *ca2b* mRNA in many tissues.

The presence of mRNA was seen in specific cells that are scattered on the yolk sac and showed that a strong signal was present from the brain to the tail bud at 14 hpf [25]. After 18 hpf, the *ca2b* mRNA was found in the specific cells on the skin of the yolk sac and yolk tube, spinal cord neurons, and pronephric duct. Later in the development, the *ca2b* expressing cells extended to the gill area in 5 dpf embryos. Identification of cell types expressing *ca2b* revealed that the *ca2b* mRNA is specifically localized to H^+^-ATPase-rich (HR) cells but not in Na^+^-K^+^-ATPase-rich cells suggesting it role in H^+^ transport in these cells [25,30].

Knockdown of *ca2b* using morpholinos showed no phenotypic defects in the morphant embryos and had no effect on the number of HR cells in the embryos [25]. The measurement of H^+^ activity on the surface of zebrafish embryos showed the highest activity (the lowest pH) at the lower part of the yolk sac in the intact embryos. However, in the morphant embryos, knockdown of *ca2b* showed no effect on the surface pH gradient at 24 hpf but reduced the surface H^+^ activity from 48 hpf compared with that of wildtype embryos [25]. In addition, knockdown of *ca2b* resulted in higher Na^+^ influx compared with control group embryos. Regulation of gene expression differed in *ca2b* morphants compared to the controls, showing showed significant increase in the expression of ca15a and nhe3b in 96hpf morphants. 

The *ca2b* was first identified using the functional genomic approach from Ensembl and NCBI databases and confirmed by cloning and sequencing the cDNA [25]. Since CO_2_ production increases during early embryonic development, it was hypothesized that CA is needed for effective excretion of CO_2_ and the pattern of CA expression shows the progress with the increase in *ca2b* mRNA [30]. Indeed, the expression analysis using RT-qPCR showed the presence of *ca2b* mRNA as early as 0 hpf and the *ca2b* mRNA was significantly higher than the ca2a mRNA at 48 hpf [30]. Determination of rates of O2 consumption and CO_2_ excretion as a function of developmental stage showed significantly higher rates at 48 hpf, coinciding with the expression of ca2b mRNA. Contribution of CA activity to CO_2_ excretion was studied using acetazolamide, a CA inhibitor and the results of the study showed significantly lower CO_2_ excretion with no effect on O_2_ consumption compared to the control group larvae. In addition, knockdown of the ca which reduced the activity of ca2b compared to the control group larvae at 48 hpf also reduced the excretion of CO_2_ significantly compared to the control group larvae [30]. The result of the study suggested that ca2b is needed for the excretion of CO_2_ during early embryonic development.

## 4. Mitochondrial *ca5* Gene Is Required for Medial Fin Development and Regulation of Acid–Base Balance during Embryogenesis

The *ca5* gene encodes for the zebrafish orthologue *CA5* and the zebrafish genome contains only one *ca5* gene for the mitochondrial CA [32], and comparison of the amino acid sequences reveals 31% identity between zfca5 and hCA VA, and 40% between zfca5 and human CAVB. Expression studies using whole-mount in situ hybridization studies showed that *ca5* mRNA is expressed in the lens and a specific part of the embryo that resembles a developing pancreas during embryonic development.

A study by Postel and Sonnenberg identified *collapse of fin* (*cof*) mutant zebrafish during a forward genetic screen from a mutant library that was created using N-Ethyl-N-nitroso-urea (ENU) mutagenesis [32]. Interestingly, the mutant larvae did not express *ca5* in the fin epidermis by in situ hybridization. The *ca5^cof^* mutant larvae showed phenotypic defects in medial fins. They had defects of epidermal integrity and the collapse of the medial fins occurred at 2 dpf. The mutant fish also exhibited cardiac failure with edema and necrosis of the yolk sac was observed later in during embryonic development. The meiotic mapping placed the *cof* allele on chromosome 25 between markers G39307 and z68140. Sequencing of open reading frames of the genes showed a T839A mutation in the coding region of the *ca5* gene. The *ca5^T839A^* mutation led to amino acid substitution of residue methionine 280 to lysine. The M280 residue is highly conserved across the species and other members of the CA protein family. The *ca5^T839A^* mutant phenotype was able to be rescued with the restoration of full length *ca5* mRNA. The study showed that the *ca5^T839A^* missense mutation that results in collapse of the medial fins. 

## 5. Carbonic Anhydrase 6 Swim Bladder Development or Function

Carbonic anhydrase VI, a secretory form of CA was originally reported as gustin by Henkin et al. (1975) from human saliva and later in 1998 it was identified as CA VI [43,44]. Several studies in humans and in knockout mice have suggested that the CA VI may have role in taste perception and immunological function [45,46,47,48,49]. However, even more than after 40 years from the first report, the precise physiological role of CA VI is still not known. As a result of sequencing non-mammalian genomes, the presence of ca6 gene was reported in zebrafish. The encoded ca6 protein was found to that contain an additional pentraxin (PTX) domain at the c-terminal end [33]. The discovery of *ca6* protein with c-terminal pentraxin domain is a unique combination among the members of CA enzyme family and as well as among the known pentraxins. The recombinant CA-PTX enzyme containing 530 amino acids showed a very high carbonate dehydratase activity. Light scattering studies showed that the ca6-PTX protein is pentameric in solution. The bioinformatic analysis suggested that the ancestral CA VI was a transmembrane protein. The exon coding for the cytoplasmic domain had been replaced by an exon coding for PTX, and later the therian lineage lost the PTX-coding, resulting in the secretory CA VI protein as we know it in mammals [33].

The immunochemical staining showed that the CA VI-PTX is expressed in the skin, heart, gills, and swim bladder with a very strong signal on the cell surfaces. The expression analysis of *ca6* mRNA using qRT-PCR showed that the gene is prominently expressed in fins/tail, and brain and low levels of expression in the gills, kidney, teeth, skin, and spleen. The knockdown of *ca6*-ptx gene using morpholinos did not showed no major morphological developmental defects during the development suggesting that *ca6* is not critical for the embryogenesis in zebrafish [33]. Interestingly, at 4-dpf the morphant embryos showed absence or deflated swim bladder [33]. The swim pattern analysis of 4-dpf morphant larvae showed decreased buoyancy and swam less efficiently compared to the control group larvae. Interestingly, when the gene expression was restored in the 5 dpf larvae, the swimming pattern also returned to almost normal. These results suggested that CA VI is required either for swim bladder development or swim bladder function. The presence of high levels of CA VI in fish and mammalian tissues allow the delivery of CA VI onto surface of physical barriers facing external environments (gut, skin, and gills in zebrafish; skin, saliva, milk, and respiratory tract in human/mouse). This finding is consistent with a proposed function associated with primary immune defense. The result from both of studies from humans and zebrafish suggest that both mammalian and fish CAVI enzymes are components of the innate immune system irrespective of the presence or absence of PTX domain. 

## 6. Carbonic Anhydrase 8 Motor Coordination

Carbonic anhydrase VIII is also known as carbonic anhydrase related protein (CARP) and is one of the members of the catalytically inactive CA isoforms [13,50,51]. The catalytic inactivity is due to the absence of one of the three histidine residues required for the enzymatic activity. Among the catalytically inactive CA isoforms, CA VIII was the first to be reported based on its expression pattern in the mouse brain [52,53]. Subsequently, several studies were carried out mainly related to its expression in both in mice and humans [13,54,55,56]. The expression studies showed that the CARP VIII is highly expressed in the cerebellum and mainly in the cerebellar Purkinje cells also during the embryonic development, suggesting a role in cerebellar development [13,54,55,56]. The most convincing evidence for the involvement of CAARP VIII in motor coordination came from studies demonstrating the spontaneously occurring mutations in the *car8*/*CA8* gene in *waddles* (*wdl*) mice and from the members of Iraqi and Saudi Arabian family [57,58,59]. The wdl mice with 19 bp deletion in *car8* gene showed wobbly side-to-side ataxic movement throughout their life span [57]. The members of Iraqi and Saudi Arabian families with a mutation in the *CA8* gene showed reduction in cerebellar volume with cerebellar ataxia and cognitive impairment, and in addition the members of Iraqi family showed quadrupedal gait [58,59].

In later studies, the bioinformatic and phylogenetic analyses showed that there is a high similarity between the vertebrate and invertebrate *CA8* sequences [13]. Zebrafish also have an ortholog called *ca8*, with unusually high similarity to the human *CA8* sequence [13]. Further analysis of zebrafish *ca8* sequence showed that it contains 9 exons that code for 281 amino acids [34]. A major part of exon 1 and small part of exon 8 and all of exon 9 encode no amino acids similar to the human transcripts [34]. The deduced amino acid sequence of zebrafish ca8 when compared with human CARP VIII showed 79% identity and the amino acid identity of the CA domain was 84%. The high degree of homology between the CARP VIII sequences suggests a conserved and essential function also in fish.

CARP VIII is known to interact with inositol 1,4,5-trisphosphate receptor type 1 (ITPR1), an ion channel protein that regulates internal *Ca2*^+^ ion release [60]. The mutational analysis using yeast two-hybrid system showed that almost the entire CA domain of CARP VIII is required for ITPR1 binding and release of intracellular *Ca2*^+^ ion [51,60]. The co-evolutionary analysis of ca8 and itpr1b from zebrafish revealed that these two proteins evolved together and the ITPR1b interacts with ca8 [51,60]. The expression analysis showed that the zebrafish *ca8* mRNA is strongly expressed in the central nervous system similar to humans [60]. Accordingly, the immunohistochemical analysis using human antibodies against the ca8 is strongly expressed in the cerebellar Purkinje cells. The expression pattern in zebrafish, both at mRNA and protein levels, suggested that the function of ca8 is conserved in zebrafish, corresponding to the function of CARP VII in mammals, and is required for both the development of cerebellum and motor coordination function [60]. The developmental expression analysis showed that presence of *ca8* mRNA at 0 hpf, suggesting that the mRNA is of maternal origin and required for the development of zebrafish brain during early embryogenesis [34]. The expression of *ca8* mRNA was also found to be high in the brain, heart, kidney, eye, and skin. Interestingly, the expression pattern of *itpr1b* mRNA was found to be similar to the expression of *ca8* mRNA, confirming that the itpr1b protein interacts with ca8 and is required during early embryogenesis along with ca8 [34].

The knockdown of *ca8* gene using gene specific translation and splice site blocking morpholinos showed abnormal changes in the head of the morphant embryos as early as 9 hpf. In addition, the *ca8* morphant larvae exhibited fragile body, curved tail, small eye size, and pericardial edema. As the development progressed the defects in the morphant larvae became more prominent and showed shortened tail, curved body axis, absence of swim bladder, and otolith vesicles [34]. The TdT-UTP nick end labeling (TUNEL) assay showed apoptosis of the cells in the head region of *ca8*-morphant larvae [34]. The transmission electron microscopy studies showed increased neuronal cell death and the apoptotic changes included a condensed nucleus, fragmented mitochondrial profiles and debris of dead cells [34]. The *ca8* morphants injected with as low as 6µM *ca8*-MOs showed ataxic movement pattern similar to humans with defective CARP VIII, suggesting the importance of *ca8* gene for brain development early during the embryogenesis, and confirms its role in the motor coordination function, similar to humans [34].

## 7. Association of Carbonic Anhydrase 9 with Intracellular Acidosis

CA IX is an extra extracellular oriented CA with a high catalytic activity and contains a single transmembrane domain spanning 25 amino acids [61,62]. Endogenously, the CA IX is expressed in the mouse skeletal muscle sarcoplasmic reticulum and epithelial cells of mammalian gastrointestinal tract [63,64,65,66]. In humans, CA IX is known to be under the control of hypoxia inducible factor (HIF) and is overexpressed in several types of cancers and has been the focus of intensive research during the last three decades [67]. Tumor cell environment is characterized by hypoxia and is responsible for overexpression of CA IX in tumors [68]. It is hypothesized that the CA IX promotes tumor growth by creating acidic extracellular environment that is required for tumor cell proliferation, spread and escape from apoptosis [68,69,70]. However, for this to happen a functional association of CA IX to a proton or bicarbonate transporter is needed, but no such association has been directly shown so far, and no activity of any bicarbonate transporter has been found to increase along with the expression of CA IX [69,70]. Therefore, the exact role of CA IX in cancer cells is not fully established. Interestingly, the proteoglycan-like domain of CA IX has been implicated in tumor metastasis, probably by interfering with normal cell adhesion mechanism [71,72,73]. However, not much is known about the physiological role of CA IX in normal tissues or its role in hypoxia response in normal tissues.

To study the role of CA IX in normal tissues in both normoxic and hypoxic environments zebrafish is an excellent model as it encounters hypoxic conditions regularly as evidenced by well-developed hypoxia responses and response systems [70,74]. Indeed, a study was conducted to examine the presence and expression of ca9 and its hypoxic response in zebrafish. Similarly, the effectiveness of hypoxia inducible CA expression in reducing intracellular acidosis was also investigated. Database search revealed the presence of hypothetical protein similar to mammalian CA IX in GenBank (XM_689890) (Figure 1). The transcript size of the putative gene contains 1560 bp with a coding region of 1155 pb. The predicted structure of the protein showed that it contains an N-terminal signal peptide (amino acids 1–22) and a single transmembrane domain (amino acids 327–349), suggesting it to be a membrane-associated CA isozyme [35]. Phylogenetic analyses grouped the sequence closely with mammalian CA IX isozymes (Figure 1). Interestingly, unlike NH2-terminal region of mammalian CA IX, in zebrafish there is no proteoglycan-like binding (MN) domain, suggesting that the predicted function of ca9 in hypoxia is due to the catalytic activity of the enzyme [61].

Expression analyses showed the presence of CA IX mRNA in the eye and gut and low levels present in the kidney, liver and brain and very low levels in muscle. There was an increase in the expression of ca9 in the brain, eye and muscle under hypoxic conditions, suggesting that ca9 shows similar hypoxic responses as previously shown in mammals. In the brain, ca9 was localized to junctions between neurons and astrocytes, and may be involved in maintaining the proper proton gradients for lactate shuttling. Genes that are transcriptionally regulated in response to hypoxia, including human CA IX, contain a gene enhancer motif termed hypoxia-responsive element (HRE), where hypoxia-inducible factor-1 (HIF-1) binds to DNA [75]. In zebrafish, multiple potential HREs were found within the zebrafish CA IX sequence mainly in various introns.

## 8. Carbonic Anhydrase 10a and 10b Is Required for Embryonic Development and Plays a Role in Motor Coordination

CARP X and CARP IX are the other two members of the catalytically inactive CA isoforms in mammals, and the catalytic inactivity is due to the absence of two and three of the possible three histidine residues, respectively, that are required for the enzymatic activity of the CAs [13]. The presence of *CA10* and *CA11* genes were first mentioned by Hewett and Tashian in 1996, based on many expressed sequence tags (ESTs) [53]. Later, the sequence of *CA10* was discovered during the screening of cDNA library from human brain [76] and CA11 was identified during the construction of a physical map for the cone-rod retinal dystrophy [77]. The expression analysis of the CA 10 gene showed the presence of *CA10* mRNA in salivary glands, kidney and brain, and the analysis in adult human and fetal tissues showed presence of significant signals in all parts of the central nervous system [76]. Similarly, immunohistochemical staining showed CARP X protein in many parts of the human brain [55,56]. The expression analysis of CA RP XI both at mRNA and protein levels first showed presence of *CA11* mRNA only in the brain. However, a more sensitive RT-PCR method, showed positive signal in the pancreas, liver, kidney, salivary glands and spinal cord [77,78]. Immunohistochemical staining with monoclonal antibodies showed presence of CA XI protein in many parts of both adult and in fetal brain [55,78]. The expression analysis studies showed that the CA X and CA XI proteins play a pivotal role in the fetal and adult brain specimens.

Analysis of CA 10 and CA 11 sequences in multiple genomes showed that the *CA11* gene emerged through a gene duplication process from *CA10* after the divergence of the fish and tetrapod lineages [13]. As a result, the zebrafish and other ray-finned fishes have two *CA10* orthologs, *ca10a* and *ca10b* [13]. The *ca10a* is highly similar to mammalian *CA10* (90% identity to human CARP X at protein level), whereas *ca10b* is slightly more diverged with 75% identity between human CARP X and zebrafish cab. The expression analyses of *ca10a* and *ca10b* genes using RT-qPCR in zebrafish showed that these genes are strongly expressed in the nervous system and also in developing embryos. The *ca10a* was highly expressed in the brain, heart and eye, whereas the highest levels of *ca10b* were found in the ovary, brain and swim bladder, and a moderate signal was found in the testis, spleen and eye. The developmental expression showed that the *ca10b* mRNA was of maternal origin and with the highest presence at 0 hpf and later at 96 hpf and remained high until 168 hpf. The expression pattern of the *ca10b* gene suggested that this gene plays an important role in the early embryogenesis and is required throughout the developmental period in zebrafish. The presence of *ca10b* was also seen throughout developmental period of zebrafish embryos. The highest expression of ca10 be seen at 96 hpf and remained high till 168 hpf. The expression pattern of *ca10a* suggested that this gene plays an important role during the embryonic development especially after 72 hpf [36]. Evolutionary conservation of *CA10*-like genes, their ubiquitous expression pattern in different tissues, and high mRNA levels during embryonic development suggests a crucial role for CARP X-like proteins in vertebrates including zebrafish [36,51].

The morphant larvae knockdown for *ca10a* gene using anti-sense morpholinos showed defects in the head and had abnormal body shape and small eyes. With the progress in development, the abnormalities became more prominent showing long curved tail, and curved body, pericardial edema, and absence of swim bladder and otolith sacs. The *ca10b* morphant larvae injected with morpholinos developed more severe phenotypic defects compared to *ca10a* morphants, and the phenotypic defects appeared as early as 12 hpf. The *ca10b* morphant larvae had a short and abnormal shaped body already at 24 hpf, the body of the larvae was very fragile, and they showed a high mortality rate [36]. The *ca10b* morphant embryos had difficulty in hatching and showed abnormal body, smaller head and eye, mild pericardial edema and absence of otolith sacs, unutilized yolk sac and curved tail, and they could not survive beyond 3 dpf. The TUNEL assay on sections of *ca10a* 5 dpf morphant zebrafish larvae showed apoptotic cells especially in the head and eye regions. Similarly, large areas of apoptotic cells were observed in the head region of 5 dpf *ca10b* morphant larvae and a weaker signal in the tail region. The *ca10a* and *ca10b* morphant embryos showed abnormal movement pattern, suggesting the association of these genes in motor coordination function in zebrafish. The *ca10a* and *ca10b* morphant larvae could be partially rescued when *ca10a* and *ca10b* MOs were co-injected with capped human mRNAs for *CA10* and *CA11* genes. The partial rescue of *ca10a* and *ca10b* morphant embryos with the injection of gene-specific human mRNAs also confirmed the specificity of the *ca10a* and *ca10b* antisense morpholinos used in the study. The phenotypes of the zebrafish larvae obtained by knockdown of *ca10a* and *ca10b* using morpholinos were also confirmed by silencing these genes using CRISPR/Cas9 genome editing technology [36]. Similar to the morpholinos injected larvae, the *ca10b* mutated larvae showed severe phenotype with high rate of mortality at 1 dpf and larvae did not survive beyond 2 dpf. The *ca10a* mutated larvae showed a less severe phenotype with a lower mortality rate at 1 dpf. The expression pattern of *ca10a* mRNA suggests that ca10a protein plays an important role in the brain, eye and several other tissues. Accordingly, ca10b might play some roles in reproduction and brain functions.

## 9. Carbonic Anhydrase 14 Play an Important Role in Melanocyte Maturation

*Carbonic anhydrase 14* cDNA clone from human was sequenced and reported to contain 1757 bp encoding a 337 amino acid polypeptide with a molecular weight of 37.6 kDa [79]. The amino acid sequence of CA XIV was shown to be highly similar to another transmembrane CA isoform, CA XII [79]. In addition to its role in acid base regulation, the CA XIV is implicated in elicitation of functional retinal light response in mice [80]. The studies related expression of CA XIV and its role in humans and mouse as discussed in the earlier, Hoek et al. reported that microphthalmia-associated transcription factor (MITF) may regulate the expression of CA XIV in melanocyte cells [81]. Indeed, the prior evidence for the regulation of CA14 expression by MITF was confirmed using *ca14* zebrafish knockdown and knockout models [37].

To demonstrate the role of CA XIV on the function of melanocytes, morpholino oligonucleotides were used to knock down the *ca14* gene expression in zebrafish. In zebrafish, the cells that produce the pigment are termed as melanophores and are functionally similar to melanocytes of higher vertebrate with conserved gene networks [37]. The *ca14* morphants at 2-dpf showed light pigmentation compared to the control group larvae. The significant reduction in pigmented melanophores in *ca14* morphants suggests that ca14 plays an important role in the process of melanogenesis, a crucial event associated with melanocyte maturation. During normal melanocytes maturation there is an increased expression of pigmentation genes tyr, dct, and tyrp1b. The *ca14* morphant fish showed reduced expression of these genes, suggesting that ca14 interferes melanocyte maturation by altering the expression of genes involved in pigmentation. Based on these results we believe that CA XIV mediates pigmentation gene expression through a transcriptional response. As CA XIV is regulated by MITF and considering its ability to control pHi and its role in sustaining melanin content of zebrafish melanophores, it is a likely candidate to mediate melanocyte maturation.

To confirm the role of ca14 on pigmentation in zebrafish, the coding region of *ca14* gene was targeted using CRISPR-Cas9 system. The mutant fish showed a frameshift mutation in the third codon of *ca14* gene by the deletion of two bases, and thus encoded a truncated protein. The *ca14* knockout mutants showed small eye size, enlarged heart, and decreased pigmentation. The pigmentation phenotype observed in the morphant embryos was recapitulated in the genetic mutants, confirming the regulatory role of CA XIV in the maturation of melanocytes. At the adult stage, the *ca14* mutation induced a visible decrease in pigmentation. The expression analysis of differentiation genes at 36 hpf, the time period when the pigment cells undergo migration and maturation showed reduced levels of tyr, tyrp1b, and dct in the mutant embryos. The down regulation of these genes confirmed that the cells are at an immature less pigmented state and that pigmentation promoting gene expression is severely reduced in the absence of ca14. Thus, the study using zebrafish as a model indicates that ca14 plays an important role in the modulation of melanocyte maturation process [37].

## 10. Carbonic Anhydrase 15a Plays a Role in Acid–Base Balance and Na^+^ Uptake

Lin et al. identified 19 CAs in fish using functional genomics approach, and of the 19 *ca* genes *ca15a* was annotated by comparing them to mammalian CAs as shown Figure 1 [25]. Expression analysis using whole-mount in situ hybridization showed the presence of *ca15a* mRNA in specific cells scattered on the yolk sac and strong presence from the brain to the tail bud at 14 hpf [25]. The *ca15a* expressing skin cells increased as the development progressed, and they extended to the gill region in 5ddf embryos. Localization studies by triple labelling studies for mRNAs of *ca15a* Na^+^, K^+^-ATPase and H^+^-ATPase showed localization of *ca15a-* positive signal in HR cells. Expression studies were also carried out using RT-PCR, and the presence of *ca15a* mRNA was mainly found in the brain, eye, and muscle and moderate presence was observed in the intestine, heart, gills and spleen.

Knockdown of *ca15a* using morpholinos produced effects on the surface H^+^ gradient in the morphants. The *ca15a* morphants showed increase in the surface H^+^ concentration at 24 hpf, which was recovered to the normal level as observed in wild type fish at 48 hpf. Studies on the effect of *ca15a* knockdown on Na^+^ uptake function showed a significantly higher Na^+^ influx compared to the control and wild type embryos.

Expression analyses of relevant genes in the ca2a morphant embryos showed that there was a significant increase in the expression of *ca15a*, and on the other hand, the ca15a knockdown embryos showed decreased expression of zebrafish V-type proton ATPase catalytic subunit A (zatp*6v1a*) and increased expression of Na^+^/H^+^ exchanger 3b (zNHE3b). The findings suggested suggests that *ca15a* is associated with both acid–base regulation and Na^+^ uptake mechanism in zebrafish HR cells [25].

Another study was carried out to clarify the mechanism of Na^+^ uptake by analyzing the expression of 12 ca isoforms and the genes associated with Na^+^ uptake by Ito et al. [27]. The expression analyses showed presence of high levels ca15a mRNA in H^+^-ATPase/mitochondrion-rich cells (H-MRCs) in the zebrafish larvae. The expression of *ca15a* mRNA was dependent on salinity of the water, whereby the *ca15a* mRNA was upregulated at 0.03 mM NA^+^ water. Immunohistochemical analysis showed localization of ca15a in the apical membrane and external surface. Knockdown of ca15a using morpholinos resulted in significant reduction in Na^+^ accumulation in H-MRCs. Colocalization studies showed that ca15a is closely associated with Na^+^/H^+^ exchanger 3b (Nhe3b) and ammonia transporter Rhcg1. Similar to the finding of Lin et al., the findings of this study showed that ca15a play a crucial role in Na^+^ uptake, and accordingly, ca15a probably forms a transport metabolon with Nhe3b for its physiological function [27].

## 11. Carbonic Anhydrase 15b Is Required for Migration of Primordial Germ Cells in Developing Embryos

Wang et al. carried out the expression analysis of ca15b using sequences (NM_213182) obtained from NCBI database [82]. The amplified cDNA sequence contained 1716 bp with a 918 bp open reading frame preceded by a 93 bp 5′-untranslated region (UTR) at N-terminal end and a 705 pb 3′ UTR. The zebrafish ca15b gene contains 9 exons and eight introns spanning 6420 bp genomic region that is located on chromosome 12 (Table 2). The *ca15b* gene encodes a 305 amino acid residue protein and contains a CA IV-XV like domain. The calculated molecular weight of the protein of the protein was 33.29 kDa, and the protein contains nine potential N-glycosylation sites and one potential O-glycosylation site. It is predicated to contain a transmembrane region in the C-terminal end of the protein (aa 280–302). The N-terminal region (aa 1–279) was predicated to be localized extracellularly and the short C-terminal tail (aa 303–305) is localized in the cytoplasm and the protein did not show any signal peptide. Phylogenetic analyses showed significant homology between ca15b is similar to the other ca15s of zebrafish and CA IV of mouse (Figure 1). Expression analysis using RT-PCR showed presence of ca15b mRNA in the ovary, heart, brain and muscle. In situ hybridization studies showed the presence of a mRNA signal throughout the cytoplasm of stage I to II oocytes and localized signal in the cortex of stage III oocytes and periphery of the stage IV oocytes. Western blot analysis used to study the expression of ca15b protein showed its expression specifically in the ovary, heart, brain and muscle, similar to the expression of its mRNA in different tissues. During the embryonic development, ca15b mRNA was found in every blastomere of the embryos from the one-cell stage to the blastula stages and was localized in the primordial germ cells (PGCs) from the two-cell stage to 24 hpf embryos. Based on the results of the expression analyses, it was concluded that ca15b is possibly involved in the development of PGCs and female germ cells in zebrafish.

In the recent past, Tarbashevich et al. showed a mechanism for cell polarization in vivo in response to chemokine signaling [38]. The study, involving zebrafish embryos, demonstrated that the cells respond to the graded distribution of the guidance cue by elevating pH at the cell front, thus promoting the Rac1 activity and actin polymerization at the leading edge of the migrating cells (Figure 2). The authors used PGCs as these cells share many properties with metastatic cancer cells, including motility initiation, mode of migration and response to chemotactic cues [84,85,86,87,88,89].

In this study, the expression analysis of ca genes showed that zebrafish PGCs express ca15b and its mRNA becomes enriched 15-fold in these cells at the time of migration onset relative to somatic cells. The *ca15b* mRNA is a maternally originated transcript expressed in the blastomeres and in the germplasm during early cleavage stages. During this period PGCs normally migrate toward the region where the gonad develops (4–24 hpf). The expression pattern of *ca15b* in this study was similar to the expression patter found in the earlier study as described above [82].

To study the role of ca15b in controlling pH changes in PGCs, the *ca15b* gene was targeted with the translation blocking morpholinos (MOs) targeting the ca15b gene. In the morphant embryos, a global reduction in pH was observed, in addition to reduced pH elevation at the front. In the ca15b morphants, 20% of the PGCs failed to reach their migration target and showed a 2-fold increase in the number of ectopic cells relative to control embryos. This was due to the defective pH elevation at the front. Interestingly, the effect of ca15b knockdown could be increased by reducing the level of the Cxcl12a, the chemokine that constitutes the guidance cue to migrating PGCs. (Figure 2). A mutation in the cxcl12a gene enhances the phenotypic effect of ca15b morphant by eliminating the polar pH distribution in PGCs. These results suggested that the main function of chemokine signaling is to initiate a cascade leading to polarized pH distribution that depends on Ca15b function, rather than to regulate the global pH in the cells.

Analysis of individual migration tracks of PGCs were carried out to study the precise role of pH elevation at the cell front in PGC migration. Interestingly, the migration tracks of ca15b morphant PGCs showed reduced straightness, a parameter suggesting how many turns are made by a cell when migrating. Comparison of the behavior and shape of manipulated cells with those of control cells was carried out to study the basis of defective migration of cells with decreased ca15b activity, this showed a significant increase in total amount of blebs in PGCs upon ca15b knockdown. In addition, there was a significant decrease in the activity of Rac1and reduced actin polymerization at the front of cells. The Rac1, a member of the Rho-GTPase family and a key regulator of actin dynamics required for the cell motility through the actin network [90]. The study showed that the polarized pH distribution depends on a polarized chemokine signal and on the function of ca15b.

## 12. Conclusions

To date, eleven different CAs have been studied using zebrafish knockdown or knockout models, which has generated a significant amount of new information about these proteins. The role of ca14 in melanocyte maturation is well-documented. Importantly, the roles of catalytically inactive CAs, CARPs, have also become more evident. Previously these proteins have not gained much attention, but, in the light of zebrafish studies, they have crucial functions especially in the central nervous system. A unique novel *CA6*-pentraxin protein has been reported in zebrafish, and its role in swim bladder function or development is obvious. According to several studies published to date, several isoforms of CAs are vital for developing embryos and larvae. Although zebrafish models have certain limitations, they can be highly valuable for phenotypic analyses. Importantly, tissues and organ systems that are difficult to in humans (e.g., central nervous system or inner ear), are available for researchers when using zebrafish as a model organism. Zebrafish models have, indeed, opened multiple new avenues for CA research, and a lot still remains to be discovered.

## Figures and Tables

**Figure 1 ijms-23-04342-f001:**
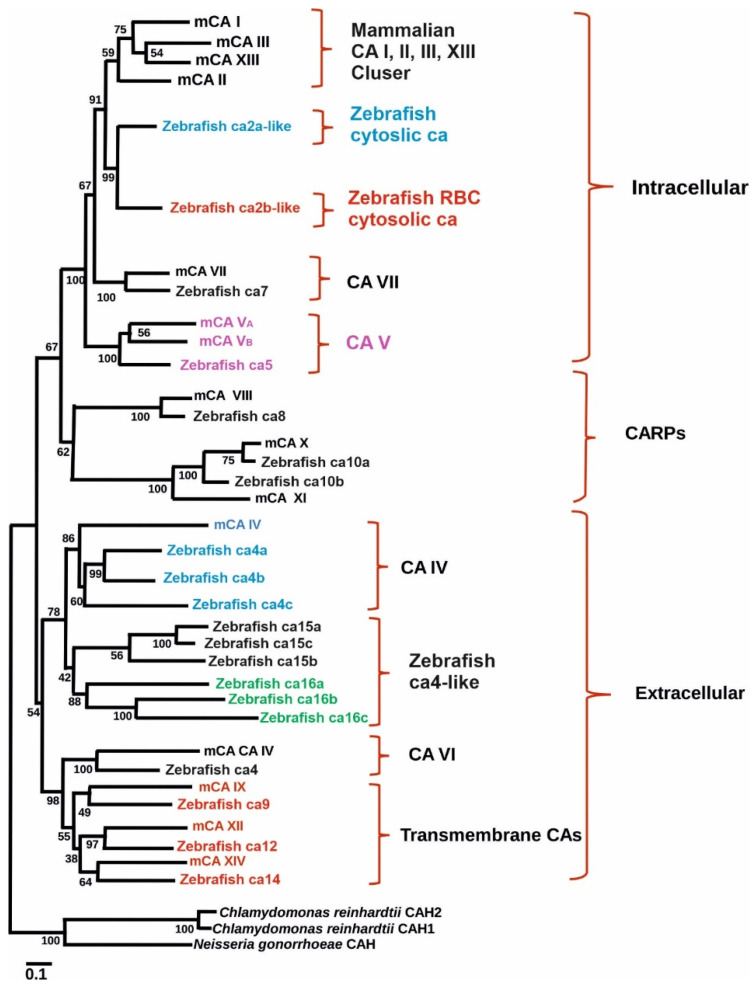
Phylogenetic tree showing the relationship between carbonic anhydrases of zebrafish with mouse carbonic anhydrase. Figure modified from [25].

**Figure 2 ijms-23-04342-f002:**
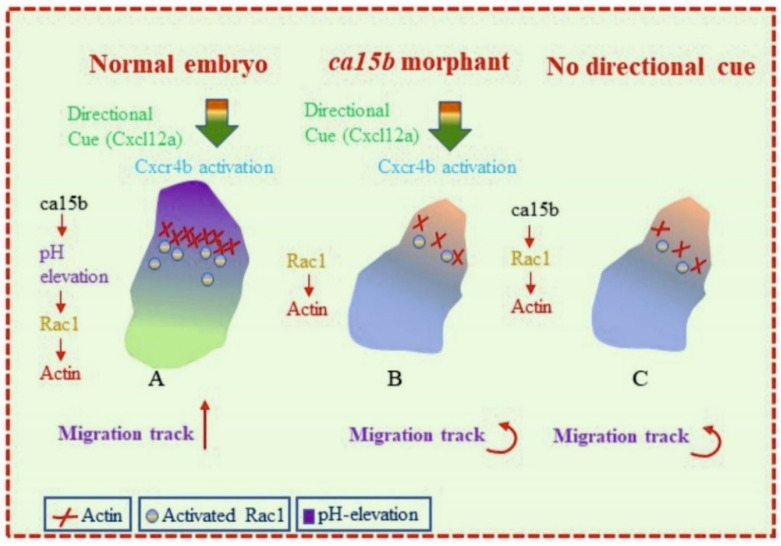
Proposed role of ca15b in the developing zebrafish embryos: (**A**) In the developing embryos, the migrating progenitor cells elevate pH due to CA enzymatic activity of ca15b at the cell front in response to chemokine signaling by C-X-C motif chemokine 12 (Cxcl12a). The increased pH at the cell front helps in maintaining the cell polarity. The elevated pH increases the activity of Rac1 and actin polymerization at the cell front. (**B**,**C**) In the absence of either ca15a activity or directional clue by Cxcl12a, no migration of the PGCs occurs in the developing zebrafish embryos. (Cxcr4b: Chemokine (C-X-C motif), receptor 4b. This protein shows a G protein-coupled chemoattractant receptor activity and plays role in several processes, including gamete generation). (The image was remade based on the graphical abstract by Tarbashevich et al. [38].

**Table 1 ijms-23-04342-t001:** The details of studies on zebrafish CAs and their proposed roles.

CA	Study	Tissues/Cells with the Highest Expression	Main Proposed Function	Ref.
*ca2a/ca17a*	Knockdown and knockout	H^+^-ATPase-rich (HR) cells	Cl-uptake, vital for zebrafish larvae	[23,24,25,26,27,28,29,30,31]
*ca2b*	Knockdown			[30]
*ca5*	Forward genetics	lens, developing pancreas	Acid-base homeostasis, medial fin and embryos development,	[32]
*ca6*	Knockdown	skin, heart, gills and swim bladder	Swim bladder development/function, primary immune defense	[33]
*ca8*	Knockdown	central nervous system (especially in Purkinje cells), heart, kidney, eye and skin	Motor coordination	[34]
*ca9*	Biochemical study	eye and gut and kidney, liver and brain	Hypoxic response	[35]
*ca10a*	Knockdown and knockout	brain, heart and eye	Embryonic development, motor coordination, vital for zebrafish larvae	[36]
*ca10b*	Knockdown and knockout	ovary, brain and swim bladder	Embryonic development, motor coordination, vital for zebrafish larvae	[36]
*ca14*	Knockdown and knockout	brain, retinal pigmented epithelium, liver, heart and skeletal muscle	melanocyte maturation	[37]
*ca15a/ca4c*	Knockdown	Skin, gills, H-ATPase-rich (HR) cells	Na^+^ uptake, acid base regulation mechanism	[25,27]
*ca15b*	Knockdown			[38]

**Table 2 ijms-23-04342-t002:** The details of zebrafish carbonic anhydrases presented in the article.

CA	IDs: Ensemble, RefSeq and or UniProt	Chromosome/Exons	Amino Acids	Location	MW (kDa)	Ref.
*ca2a/ca17a*	NM_199215.2/ENSDART00000038364.9	Chr24/7	260	Cytoplasmic	28.91	[23,24,25,26,27,28,29,30,31,83]
*ca2b*	ENSDART00000013411.6/NM_131110	Chr2/7	260	Cytoplasmic	28.68	[30]
*ca5*	ENSDART00000162750.2/NM_001111201.1/A8KB74	Chr 25/7	310	Mitochondrial	35.62	[32]
*ca6*	ENSDART00000132733.3/XM_002666479.3	Chr 23/9	530	Secreted	60.2	[33]
*ca8*	ENSDART00000140012.2	Chr 2/9	281	Cytoplasmic	32.08	[34]
*ca9*	ENSDART00000168549.2/XM_689890.8/A0A0R4IHP7	Chr 10/11	395	Transmembrane	44.08	[35]
*ca10a*	F1QMF0/ENSDART00000178562.2	Chr12/10	328	Secreted	37.5	[36]
*ca10b*	ENSDART00000055264.7/E7EYF2/XM_005164096.3	Chr3/9	326	Secreted	37.72	[36]
*ca14*	ENSDART00000149574.2/NM_001328144.1/F8W4I7	Chr16/12	373	Transmembrane	41.57	[37]
*ca15a/ca4c*	CN509883/ENSDART00000008893.10/F1Q816 XM_017358460.1	Chr 12/9	324	Transmembrane	34.86	[25,27]
*ca15b*	ENSDART00000152521.3/NM_213182/R4GDY8	Chr 12/9	320	Transmembrane	35.05	[38,82]

## Data Availability

Not applicable.

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
