# Peer review of "Roles of Carbonic Anhydrases and Carbonic Anhydrase Related Proteins in Zebrafish"

_ijms, 2022, doi:10.3390/ijms23084342_

Round 1
Reviewer 1 Report
Manuscript Number: ijms-1651172
entitled: Roles of carbonic anhydrases and carbonic anhydrases related proteins in zebrafish
This is an interesting scientific study. Therefore, the manuscript is suitable for publication in the International Journal of Molecular Sciences after considering the below comments:
- Would you please add the range of years in the abstract part?
Reviewer 2 Report
The article entitled “Roles of carbonic anhydrases and carbonic anhydrases related proteins in zebrafish" reviews the current knowledge of CA-related studies performed in zebrafish Although the paper provides a fairly well-structured, well-summarized study of CA-related studies in the zebrafish model system relevant to drug discovery, I suggest a number of modifications and improvements to the manuscript. . My comments on this are summarized below. The general introduction gives a well-structured and appropriate overview of the topic. Respectively, the CA-related topic points also well describe the pivotal points of the model. However, 2.-10. points, the authors are too immersed in the presentation and the relevant information for the non-expert reader is already lost. This is somewhat remedied by Table1 and Table2 and Fig1, but even so, the essential information is difficult to follow. In the context of the above, I would suggest presenting the information at each point in a simplified, more accessible form, focusing on meeting the information needs of early-stage drug research and modeling.
Some suggestions for minor corrections:
- line 22: ‘carbon dioxide according to the following equation:’ please correct to carbon dioxide according to the following equation of bicarbonate equilibrium:
- line30: ‘α-CAs are the best characterized CA class, and the enzymes…’ please correct to α-CAs belonging to CA class are the best characterized, and the enzymes…
- line 37: ‘some are catalytically inactive and called…’ please correct to some of them are catalytically inactive and called…
- in the context of line 46-49: I would see a need to demonstrate the use of zebrafish in HTS systems. In connection with typical examples, criteria, limitations.
- line 165: ‘gene for the mitochondrial CA [37]. and comparison of…’ please correct to gene for the mitochondrial CA [37], and comparison of…
- line 267: ‘he knockdown of ca8 gene using gene…’ please correct to The knockdown of ca8 gene using gene
Round 2
Reviewer 2 Report
The manuscript is acceptable in the present form.